# BED-YOLO: An Enhanced YOLOv10n-Based Tomato Leaf Disease Detection Algorithm

**DOI:** 10.3390/s25092882

**Published:** 2025-05-02

**Authors:** Qing Wang, Ning Yan, Yasen Qin, Xuedong Zhang, Xu Li

**Affiliations:** 1College of Information Engineering, Tarim University, Alaer 843300, China; tlm121133@163.com (Q.W.); 18295549487@163.com (N.Y.); qys_1738828465@163.com (Y.Q.); 2Key Laboratory of Tarim Oasis Agriculture, Ministry of Education, Tarim University, Alaer 843300, China

**Keywords:** deep learning, object detection, YOLOv10, tomato, disease detection

## Abstract

As an important economic crop, tomato is highly susceptible to diseases that, if not promptly managed, can severely impact yield and quality, leading to significant economic losses. Traditional diagnostic methods rely on expert visual inspection, which is not only laborious but also prone to subjective bias. In recent years, object detection algorithms have gained widespread application in tomato disease detection due to their efficiency and accuracy, providing reliable technical support for crop disease identification. In this paper, we propose an improved tomato leaf disease detection method based on the YOLOv10n algorithm, named BED-YOLO. We constructed an image dataset containing four common tomato diseases (early blight, late blight, leaf mold, and septoria leaf spot), with 65% of the images sourced from field collections in natural environments, and the remainder obtained from the publicly available PlantVillage dataset. All images were annotated with bounding boxes, and the class distribution was relatively balanced to ensure the stability of training and the fairness of evaluation. First, we introduced a Deformable Convolutional Network (DCN) to replace the conventional convolution in the YOLOv10n backbone network, enhancing the model’s adaptability to overlapping leaves, occlusions, and blurred lesion edges. Second, we incorporated a Bidirectional Feature Pyramid Network (BiFPN) on top of the FPN + PAN structure to optimize feature fusion and improve the extraction of small disease regions, thereby enhancing the detection accuracy for small lesion targets. Lastly, the Efficient Multi-Scale Attention (EMA) mechanism was integrated into the C2f module to enhance feature fusion, effectively focusing on disease regions while reducing background noise and ensuring the integrity of disease features in multi-scale fusion. The experimental results demonstrated that the improved BED-YOLO model achieved significant performance improvements compared to the original model. Precision increased from 85.1% to 87.2%, recall from 86.3% to 89.1%, and mean average precision (mAP) from 87.4% to 91.3%. Therefore, the improved BED-YOLO model demonstrated significant enhancements in detection accuracy, recall ability, and overall robustness. Notably, it exhibited stronger practical applicability, particularly in image testing under natural field conditions, making it highly suitable for intelligent disease monitoring tasks in large-scale agricultural scenarios.

## 1. Introduction

Tomato is one of the most economically valuable vegetables globally, serving both as a food source and an ornamental crop [1,2,3]. In recent years, the cultivation area of tomatoes has expanded continuously, making it one of the highest-yielding vegetable crops worldwide, with consumption ranking first among all vegetables [4,5]. As a crucial economic crop in various countries, tomato production not only serves as a key source of income for farmers but also plays a vital role in enhancing agricultural efficiency [6]. However, tomato diseases remain the primary constraint affecting yield and quality, with varying degrees of occurrence across different regions and encompassing diverse diseases such as early blight, late blight, leaf mold, and target spot [7]. The rapid and accurate identification and control of tomato diseases can significantly reduce economic losses and ensure the sustainable development of the tomato industry [8,9,10,11]. Therefore, the research and prevention of tomato diseases are of great importance.

Traditional methods for detecting tomato leaf diseases primarily rely on manual observation and expert judgment. However, due to the diversity and complexity of diseases, manual diagnosis is often costly, subjective, and lacks accuracy [12,13]. With the rapid advancement of computer technology, the application of object detection technology in tomato disease monitoring and prevention has gradually become a crucial direction for modernizing tomato production [14,15]. Deep learning, with its powerful autonomous learning capabilities, has significantly improved neural network performance and emerged as a vital trend and innovation in agricultural disease detection [16,17,18]. Compared with traditional machine learning methods, deep learning exhibits remarkable advantages in object detection algorithms [19,20,21], offering faster detection speed and higher accuracy, and enabling the precise identification of different types of diseases [22,23,24].

Recent advances in plant disease detection have led to the development of numerous deep learning-based models aimed at enhancing accuracy and robustness. For instance, Xie et al. integrated Inception-v1 and SE modules into a GLDD and Faster R-CNN-based detection framework, achieving a precision of 81.1% mAP and a detection speed of 15.01 FPS on the GLDD dataset—sufficient for real-time grape disease monitoring. The model demonstrated the capacity to identify multiple lesions of the same disease on a single leaf and to detect multiple distinct diseases concurrently [25]. In comparison, single-stage detectors have shown broader applicability in plant disease detection tasks due to their superior inference speed and competitive accuracy [26,27,28]. Among them, the YOLO series has emerged as a dominant solution owing to its efficiency and real-time capabilities [29,30,31]. Continuous efforts have been made to optimize the YOLO architecture for agricultural applications [32,33]. For example, Lv et al. enhanced YOLOv5 by integrating an attention mechanism and a Transformer encoder, thereby improving its discriminative power for apple leaf diseases. Their improved YOLOv5-CBAM-C3TR model excelled in distinguishing visually similar diseases such as Alternaria blotch and gray spot [34]. Similarly, Yang et al. addressed the challenges of field environments by proposing a YOLOv8-based method for corn leaf disease detection. By constructing a real-world image dataset and incorporating a Slim-neck design with a GAM attention module, their model achieved accurate detection under complex backgrounds, supporting intelligent agricultural disease monitoring [35]. In recent years, deep learning approaches have also been widely explored for tomato leaf disease detection. Liu et al. proposed an improved YOLOX model incorporating MobileNetV3 as the backbone for lightweight feature extraction, and introduced a CBAM attention module between the backbone and neck to enhance sensitivity to lesion regions. Their approach significantly improved detection accuracy while reducing memory usage and inference latency [36]. Wang et al. developed a model named TomatoDet, specifically targeting real-world challenges such as image noise and small-spot detection under field conditions. By embedding Swin-DDETR’s attention mechanism into the backbone and adopting a Meta-ACON dynamic activation function, they strengthened the model’s capacity to capture fine-grained features. The results confirmed TomatoDet’s superior precision and detection efficiency compared to baseline models, highlighting its practical potential [37]. Additionally, Abulizi et al. utilized YOLOv9 as a base model and introduced DySample, a lightweight dynamic upsampling mechanism, to improve the detection of small lesions and suppress background interference. They further enhanced boundary localization by incorporating the MPDIoU loss function, achieving over 2% improvement across multiple metrics [38].

Despite these advances, several limitations persist in existing plant disease detection methods. Many models depend heavily on curated laboratory datasets and exhibit limited generalization under real-world environmental variability. Others, while excelling at small lesion detection, often compromise computational efficiency or lack deployment flexibility [39,40]. These shortcomings are particularly pronounced in the context of tomato leaf disease detection in natural settings. Tomato foliage often exhibits overlapping leaves, occlusions, intertwined stems, and fluctuating lighting conditions, complicating the extraction of lesion features. Moreover, the small size and subtle morphology of tomato leaf spots—especially those with indistinct patterns—pose further detection challenges. The complex background can also obscure disease features and reduce detection precision. These challenges necessitate the development of a detection framework with enhanced small object recognition, robust background suppression, and high deployability. Given its efficiency and modular flexibility, the YOLO series has become a cornerstone in agricultural disease detection. The recently introduced YOLOv10n adopts a unified label assignment strategy and eliminates the traditional non-maximum suppression (NMS) module, offering increased adaptability for plug-and-play architectural optimization. These attributes make YOLOv10n a suitable foundation for building a lightweight, high-speed detection model tailored for small object detection.

In contrast, although YOLOv11 demonstrates superior performance in general object detection tasks, its application in specialized domains such as agricultural disease detection remains limited, with few empirical studies addressing its effectiveness in fine-grained recognition tasks involving small plant lesions [41]. Moreover, YOLOv11 features a more complex overall architecture, demands substantially higher computational resources for training, and currently lacks a stable open-source implementation, thereby increasing the challenges associated with model integration and deployment [42]. In comparison, YOLOv10n offers advantages of structural stability, flexible module integration, and high inference efficiency [43], making it particularly well suited for collaborative optimization with modules such as DCN, BiFPN, and EMA proposed in this study. Accordingly, YOLOv10n was selected as the base model. Building on this foundation, to address the potential negative impact on small object detection accuracy caused by YOLOv10n’s elimination of the non-maximum suppression (NMS) step, we introduced the BiFPN module to enhance multi-scale feature fusion. Simultaneously, the EMA attention mechanism was incorporated to strengthen the model’s focus on diseased regions, thereby improving the detection performance for small lesions. Experimental results confirm that the proposed BED-YOLO model not only maintains high inference efficiency but also significantly enhances the detection of small disease spots under complex natural backgrounds, demonstrating strong potential for practical deployment in agricultural scenarios. The core contributions of the BED-YOLO model proposed in this study are as follows:(1)Introducing Deformable Convolutional Networks (DCNs) into the backbone network to replace conventional convolution enhances the model’s adaptability to leaf overlap, occlusion, and morphological variations in disease regions. This improves feature extraction from diseased areas.(2)To tackle the challenge of detecting small lesion targets, the model integrates the Bidirectional Feature Pyramid Network (BiFPN) with the FPN + PAN structure to optimize feature fusion. This approach strengthens the representation of disease regions at different scales and improves the detection accuracy of small lesions.(3)Incorporating the Efficient Multi-Scale Attention (EMA) mechanism into the C2f module within the neck part of the model enhances feature extraction effectiveness, allowing the model to better focus on diseased areas, reduce background noise interference, and ensure the full retention of disease features during multi-scale fusion.

## 2. Data Collection and Processing

### Dataset Construction and Preprocessing

The tomato leaf disease image dataset used in this study consists of two parts. The majority of the data were collected under the guidance of experts in the field at a tomato plantation base in Wujiaqu City, Xinjiang Production and Construction Corps. These images were captured in a natural environment under varying lighting conditions, including morning, noon, and evening, to ensure data diversity. Field images were captured using an iPhone 14 smartphone at a resolution of 4032 × 3024 pixels, ensuring high fidelity and detailed visual representation. Additionally, the dataset includes leaf images from different stages of disease progression, such as early-stage lesions, mid-stage infections, and severely infected leaves. Furthermore, images containing backgrounds like soil, weeds, and other vegetation were incorporated to enhance the model’s adaptability to complex environments. Samples representing different disease stages, as well as weed and soil backgrounds, are shown in Figure 1.

Given that the PlantVillage dataset was captured under controlled laboratory conditions, featuring predominantly single-leaf images with relatively low resolution (256 × 256 pixels), its utility in object detection tasks remains limited. To mitigate potential impacts on detection performance, only high-quality images with clearly delineated disease regions were selected during dataset integration. These selected samples were resampled to match the input resolution required by the YOLOv10n model. Although primarily designed for classification tasks, the PlantVillage dataset was employed in this study as a supplementary source to enhance the model’s discriminative capability across disease categories. The majority of training data were derived from field-collected images, with PlantVillage samples constituting approximately 35% of the dataset. Importantly, these images were excluded from critical evaluations involving small lesion detection, thereby minimizing their influence on performance metrics. The PlantVillage dataset is a widely used public benchmark for plant disease recognition and classification research. It encompasses 14 different plant species and 38 distinct disease categories, capturing a broad spectrum of plant pathologies. Owing to its standardized laboratory setting and relatively uniform background, the dataset minimizes environmental variability, making it particularly well suited for controlled studies focused on plant disease classification and recognition.

A total of 1874 images were collected for this study, including 1218 images obtained through field sampling in Wujiaqu City, Xinjiang, and 656 images selected from the publicly available PlantVillage dataset. To ensure the objectivity of the experimental results and enhance the model’s generalization ability, standardized preprocessing was applied to both datasets during integration. Moreover, representation for each disease type was balanced across the two sources. The four targeted diseases—early blight, late blight, leaf mold, and septoria leaf spot—were distributed approximately evenly across the combined dataset, with each category comprising a comparable proportion of the total images. Representative samples of the four disease categories are shown in Figure 2. Finally, all images were randomly divided into training and test sets at a ratio of 8:2, ensuring balanced category distribution across both sets. No duplicate samples were included, thereby guaranteeing fairness in model training and evaluation.

The precise annotation of disease regions is pivotal to effective model training in the context of tomato leaf disease detection. In this study, disease regions within each image were manually annotated using the LabelImg tool. The annotation process followed a rigorous protocol: First, domain experts with experience in agricultural disease identification reviewed and identified the disease type and location in each image. Then, bounding boxes were drawn around each lesion using LabelImg, and the corresponding disease category was labeled. All annotations were exported in YOLO format (TXT files), which included normalized coordinates and category identifiers. The entire annotation workflow adhered to a unified standard to ensure consistency and high data quality. An example of the annotation interface is shown in Figure 3.

In object detection tasks, insufficient training data can easily lead to overfitting, thereby compromising model generalization. To address this issue and enhance the diversity of the training dataset, a range of offline data augmentation techniques were applied to the curated original images. These included image rotation, random cropping, brightness adjustment, and noise injection. The primary objective of these augmentations was to expand the diversity of the training samples, thus improving the model’s robustness. Specifically, augmented images were generated for each disease category to compensate for minor class imbalances within the dataset and to strengthen the model’s learning resilience. Regarding the original data distribution, the four disease categories—early blight, late blight, leaf mold, and septoria leaf spot—were relatively balanced across both the field-collected and PlantVillage datasets, with approximately 300 to 500 samples per class. After augmentation, the total number of images was effectively doubled, and the augmented dataset was thoroughly shuffled prior to the training–test split. These augmentation strategies not only enriched the dataset but also substantially enhanced the model’s ability to adapt to varying real-world scenarios. Representative examples of the augmented images are presented in Figure 4.

## 3. Methodology and Design

### 3.1. Overview of the YOLOv10 Network

YOLOv10 represents a comprehensive advancement over YOLOv8, with extensive optimizations specifically tailored for real-time end-to-end object detection. Although it retains the classic three-part architecture—backbone, neck, and head—shared with YOLOv8, YOLOv10 introduces substantial improvements across several critical aspects, including network design, module innovation, optimization strategies, and data augmentation techniques, ultimately achieving notable gains in both detection accuracy and inference speed [44,45]. The backbone network incorporates the C2f module, which enhances multi-scale feature integration across different stages while significantly reducing computational overhead. This design improves the model’s ability to detect targets of varying scales while optimizing for both speed and accuracy. The overall YOLOv10 architecture and the structure of the C2f module are depicted in Figure 5 and Figure 6.

Traditional YOLO models depend on non-maximum suppression (NMS) to eliminate redundant bounding boxes during object detection. NMS functions by retaining the bounding box with the highest confidence while suppressing others with high overlap, effectively reducing duplication. However, despite its utility, NMS introduces considerable computational cost and inference latency, and its performance is often sensitive to hyperparameter selection [46]. To address this limitation, YOLOv10 proposes a novel NMS-free training paradigm. This strategy utilizes a unified dual-label assignment approach that integrates both many-to-one and one-to-one label matching mechanisms, thereby eliminating the need for NMS-based post-processing entirely [47]. During training, the many-to-one label assignment enriches supervisory signals, promoting comprehensive learning. In contrast, the inference stage employs one-to-one matching to directly generate final predictions, suppressing redundant boxes and significantly reducing inference time, while still maintaining high detection accuracy.

Beyond the NMS-free framework, YOLOv10 introduces systematic optimizations across its architectural components to further elevate efficiency and precision. Its design philosophy emphasizes the coordinated integration of lightweight modules and high-performance computing strategies, aiming for a holistic balance between accuracy and speed [48]. For example, the backbone structure is refined to reduce unnecessary computation, while the neck and head components adopt lightweight designs that minimize parameter count and complexity, further enhancing real-time inference efficiency [49]. The architecture of YOLOv10 exemplifies an optimal trade-off between performance and computational demand. Innovations such as a streamlined classification head and an improved spatial-channel decoupling downsampling mechanism drastically reduce complexity, ensuring that the model maintains high throughput and accuracy, even when applied to high-resolution imagery in real-time detection scenarios.

### 3.2. Design of BED-YOLO

The conventional YOLOv10n model primarily relies on traditional convolution with fixed kernels during feature extraction. This approach may lead to insufficient attention to overlapping and occluded diseased areas, as well as limited capability in capturing the blurred edges of lesions. Moreover, tomato leaf spots are typically small and exhibit indistinct morphological characteristics. During the model’s upsampling and downsampling process, some fine details may be lost, thereby affecting detection accuracy. To enhance the model’s adaptability to leaf occlusion and disease region features in complex environments, this paper proposes an efficient disease detection network, BED-YOLO, based on YOLOv10n.

First, dynamic convolution (DCN) is integrated into the backbone network to replace some conventional convolution operations, thereby enhancing the model’s perception of overlapping, occluded, and blurred-edge disease regions. To more accurately capture small disease region features, the feature fusion method is optimized by introducing a Bidirectional Feature Pyramid Network (BiFPN), which improves the detection accuracy of small lesion targets. Additionally, the Exponential Moving Average (EMA) attention mechanism is incorporated into the C2f module of the neck to enhance the feature representation of disease regions. This mechanism adaptively adjusts feature weights, reduces background noise interference, and ensures that disease information is not diluted during multi-scale feature fusion, thereby improving detection stability in complex environments. The network architecture of BED-YOLO is shown in Figure 7.

This study explores the integration of several state-of-the-art architectural modules—namely, Deformable Convolutional Networks (DCNs), Bidirectional Feature Pyramid Networks (BiFPNs), and Exponential Moving Average (EMA)—into the YOLOv10n framework, aligning with prevailing trends in deep learning-based model optimization. Similar combinations of these modules have been investigated in other domains of object detection. For example, Zhou et al. recently applied a comparable strategy in the context of infrastructure surface damage recognition, demonstrating its effectiveness in identifying structural cracks and other engineering defects [50]. However, while differences in application scenarios and design objectives do exist, they highlight the specific challenges addressed by our proposed approach. Unlike previous studies focusing on structural crack detection and three-dimensional visualization in engineering domains, our work targets small-object disease detection in complex agricultural environments, emphasizing lightweight deployment and robustness. These distinctions frame the motivation and contribution of our study within the context of plant disease detection.

#### 3.2.1. Deformable Convolution Network (DCN)

In natural environments, the morphology of diseased tomato leaves and lesions varies greatly, often complicated by leaf overlap and occlusion, making accurate recognition challenging. These conditions can lead to the loss of critical disease features and even result in false detections. Additionally, the edges of diseased regions are often blurred and exhibit irregular shapes, such as curling and jagged patterns. Variations in lighting and complex backgrounds further hinder the model’s ability to accurately localize disease areas. These factors collectively impact the performance of the original YOLOv10 model in detecting tomato leaf diseases, leading to the inadequate extraction of key features and reduced recognition accuracy.

The backbone network of the original YOLOv10 model employs traditional convolution, whose fixed kernel weights cannot adapt to the content of the input data during feature extraction. This limitation makes it difficult for the model to effectively capture fine details when dealing with disease images with complex natural backgrounds, especially in cases of leaf overlap and blurred disease edges. To address this issue, this study introduces Deformable Convolution Network (DCN) into the YOLOv10 backbone to replace conventional convolution. Figure 8 and Figure 9 illustrate the advantages of DCN over traditional convolution kernels, making it more suitable for detecting objects of varying sizes, shapes, and scales. As illustrated in Figure 8, standard convolution operations rely on a fixed, grid-like sampling pattern that distributes sampling points uniformly across the input, making them inherently unable to adapt to the shape or structure of the target. In contrast, deformable convolution introduces learnable offsets to the sampling locations, allowing the convolutional kernel to dynamically shift toward more discriminative regions, such as lesion edges or irregular surface contours. This adaptive mechanism significantly enhances the model’s capacity to represent complex edge geometries and substantially improves detection accuracy under challenging conditions, including overlapping leaves, blurred boundaries, and irregular lesion morphologies. Through this improvement, the model can more accurately capture the morphological features of diseased regions, effectively reducing false positives and missed detections caused by leaf occlusion, lighting variations, and background interference, thereby improving overall detection performance.

The principle of deformable convolution relies on a network that learns offset values, allowing the sampling points of the convolution kernel to shift on the input feature map, thereby focusing on the target region or areas of interest. Based on the original formula, deformable convolution introduces an offset for each sampling point, which is generated by an additional convolution layer from the input feature map, typically as a fractional value. The offset in deformable convolution refers to the spatial adjustment applied to standard sampling locations during the convolution process. These offsets are generated from the input feature map via an additional convolutional layer and are represented as two-dimensional vectors, typically in floating-point format. This enables sub-pixel level dynamic sampling and enhances the model’s adaptability to irregular structures. Building upon the standard convolutional formulation, deformable convolution introduces spatial displacement information at each sampling point, derived directly from the input features. These displacements, computed through an auxiliary convolutional operation, are expressed as continuous two-dimensional vectors, allowing each sampling location to undergo fine-grained adjustments based on the underlying content of the feature map. The final convolution operation then retrieves feature values at these adjusted locations via interpolation, thereby achieving flexible and structure-aware feature extraction. Figure 10 presents the network architecture of deformable convolution. As shown, the offset is generated through an additional convolution layer, which is distinct from the convolution layer that performs the final convolution operation. The “N” in the figure denotes the size of the convolution kernel’s region. In traditional convolution, the sampling area on the input feature map is a fixed square region, whereas in deformable convolution, the sampling area consists of a series of dynamically positioned points, which is the key distinction from standard convolution [51].

#### 3.2.2. Multi-Scale Feature Fusion in the Neck

YOLOv10 employs a hybrid FPN + PAN architecture within its neck module to enhance multi-scale feature representation. The Feature Pyramid Network (FPN) facilitates the extraction of information from feature maps at varying resolutions, thereby strengthening the model’s capacity to detect objects of different sizes. Specifically, FPN adopts a top–down pathway that progressively transmits high-level semantic information to lower-resolution feature maps. Through lateral connections, it fuses semantically enriched features with shallow spatial details, resulting in high-resolution feature maps imbued with strong semantic content. This structure is particularly effective in improving sensitivity to small objects across scales. The Path Aggregation Network (PAN) complements this design by aggregating features across different hierarchical levels, enhancing bidirectional feature flow—especially critical for small object detection. PAN introduces a bottom–up pathway on top of the FPN structure, reinforcing the expressive capacity of low-level features. This allows precise positional information from the lower layers to be propagated upward, supporting more accurate localization. By enabling this bidirectional information flow, PAN significantly improves inter-layer feature interaction, making it especially advantageous for identifying small and visually ambiguous targets.

This combination is well suited for object detection tasks involving targets of different sizes, thereby improving detection accuracy. However, tomato leaves in natural environments often suffer from diverse diseases with complex distributions, which are further influenced by environmental factors. When the lesions occupy a small leaf area and exhibit indistinct morphological features, higher demands are placed on the model’s feature extraction and information fusion capabilities. The FPN and PAN architecture is illustrated in Figure 11.

Although the FPN + PAN combination captures the positional information of small tomato leaf lesions through its bottom–up structure and enhances semantic understanding for larger objects, it still has room for improvement. First, the FPN + PAN architecture does not fully exploit high-scale feature maps, potentially missing critical leaf disease features and leading to insufficient detection accuracy. Moreover, during upsampling and downsampling, some feature map information may be lost, reducing feature reuse efficiency. This issue is particularly pronounced when detecting small leaf lesions, as small targets have fewer features, and losing details during multiple sampling stages can negatively impact final detection performance. To address these shortcomings in multi-scale feature extraction, especially in capturing small target features, this study introduces a Bidirectional Feature Pyramid Network (BiFPN).

BiFPN enhances the expression of features at different scales through a bidirectional feature fusion mechanism, thereby improving the model’s ability to recognize small lesions in complex backgrounds. It is particularly suitable for multi-scale, multi-level target detection tasks. By incorporating learnable weights, BiFPN can dynamically adjust the fusion of feature maps at different scales, effectively mitigating information loss and redundancy. Its bidirectional flow mechanism allows information to flow not only from low-resolution to high-resolution but also in the reverse direction, enhancing multi-scale information integration. The BiFPN architecture is depicted in Figure 12. BiFPN further optimizes feature fusion through horizontal and vertical connections, enabling the network to better handle scale variation and target occlusion. The flexibility and efficiency of this design have demonstrated robust performance and accuracy in various computer vision tasks, especially in multi-scale object detection [52].

#### 3.2.3. EMA Attention Mechanism

The detection of tomato leaf diseases is often hampered by complex backgrounds and the uneven distribution of diseased regions, making it challenging for detection models to accurately extract key features of the affected areas. Furthermore, during multi-scale feature fusion, the transmission of information between feature layers may introduce background noise, thereby weakening the expression of disease-related features and reducing detection accuracy. To address this issue, this study integrates the Exponential Moving Average (EMA) attention mechanism into the C2f module of the neck structure to enhance the effectiveness of feature fusion. By constructing a global information modeling mechanism, EMA allows the model to adaptively adjust the weights of features at different scales, strengthening the expression of diseased regions and effectively suppressing background noise interference. Compared to traditional attention mechanisms, EMA exhibits superior global perception capabilities, enabling the precise localization of diseased areas across feature maps of different scales and preventing the dilution of critical disease features during multi-scale fusion. This improvement not only enhances the representation of diseased regions but also boosts the model’s stability and robustness in complex environments, thereby improving the accuracy of tomato leaf disease detection.

The EMA attention mechanism adopts a unique three-path parallel structure, consisting of two 1 × 1 convolution branches and one 3 × 3 convolution branch. The 1 × 1 convolution branch captures global information, while the 3 × 3 convolution branch focuses on extracting local features. This design allows the mechanism to effectively extract information from different scales, thereby enhancing the feature map’s expressiveness. In the channel dimension, the EMA mechanism models the dependencies between channels through cross-channel interaction, which not only improves the understanding of feature relationships but also reduces computational complexity. To further enhance performance, EMA leverages 2D global average pooling to encode the global information extracted by the 1 × 1 branch and matches the dimensions of the 3 × 3 branch’s output for effective multi-scale feature fusion. During this process, EMA can simultaneously process global and local information, providing the model with richer contextual information. Additionally, EMA effectively models long-range dependencies and accurately embeds positional information into the feature map, enhancing the perception of target regions. Through these innovative designs, the EMA mechanism generates more precise pixel-level attention maps across multiple scales, significantly improving object detection accuracy and performance [53]. The architecture of the EMA mechanism is illustrated in Figure 13.

## 4. Experimental Results and Analysis

### 4.1. Experimental Environment

The experimental environment was based on the Windows 10 operating system, utilizing Python 3.11 as the development language. The deep learning framework employed was PyTorch 2.0.0, with CUDA version 11.8. The hardware configuration included an NVIDIA GeForce RTX 4060 graphics card (Santa Clara, CA, USA) and an Intel Core i5-12400F processor (Santa Clara, CA, USA). PyCharm was used as the development tool to ensure efficient code writing and debugging. The experimental environment is shown in Table 1. During the construction and training of the deep learning model, the training strategy and hyperparameter settings played a crucial role in the model’s performance and detection effectiveness. The training configuration for this experiment was as follows: 300 iterations, an initial learning rate of 0.01, the optimizer set to SGD, and a batch size of 16. To enhance training stability, a weight decay of 0.005 and a momentum parameter of 0.937 were applied to optimize the gradient update process.

### 4.2. Experimental Evaluation Metrics

In this study, precision (P), recall (R), and mean average precision (mAP50) were employed as evaluation metrics to comprehensively assess the performance of the tomato leaf disease detection model. These metrics effectively measure the accuracy, completeness, and overall recognition performance of the model.

Precision (P): Precision reflects the proportion of true positive targets among the predicted results. A higher precision indicates fewer false positives and higher quality in detection results. The precision formula is shown in Equation (1), where *TP* represents the number of correctly predicted targets, and *FP* represents the number of incorrectly predicted targets. By calculating precision, we can understand the reliability of the model’s predictions. A high precision implies fewer false alarms but may compromise recall.(1)$$Precision=\frac{TP}{TP+FP}\times 100\%$$

Recall (R): Recall is a critical metric for assessing the completeness of the detection model, reflecting the proportion of actual targets that are successfully detected. A higher recall indicates that the model can detect more targets, ensuring the completeness of detection results. The recall formula is shown in Equation (2), where TP is the same as above, and FN represents the number of actual targets that were not detected. Precision and recall often exhibit a trade-off: improving precision may reduce recall, and vice versa. Therefore, researchers need to balance these two metrics based on the specific task requirements to achieve optimal detection performance.(2)$$Recall=\frac{TP}{TP+FN}\times 100\%$$

Mean Average Precision (mAP50): Mean average precision is a crucial metric for evaluating model performance in object detection tasks. mAP50 indicates the average precision across all categories at an IoU threshold of 0.5. The mAP50 formula is shown in Equation (3). It reflects the model’s accuracy in recognizing targets and its effectiveness in reducing false detections and missed targets.(3)$$mAP=\frac{\sum_{i=1}^{k}\;AP_{i}}{k}$$

### 4.3. Performance Evaluation of the Improved Model

To comprehensively assess the detection performance of the improved BED-YOLO model, a comparative analysis was conducted against the original YOLOv10n model. Figure 14 illustrates the variation curves of the mean average precision (mAP50) and recall rate for both models. From the perspective of the key metric mAP50, the improved BED-YOLO model demonstrates a more stable convergence trend during training, with reduced fluctuation in the precision curve and significantly enhanced stability. Notably, the BED-YOLO model achieved an mAP50 of 91.3%, significantly outperforming the original YOLOv10n model across various IoU thresholds. Furthermore, in terms of recall rate, the BED-YOLO model effectively reduced the missed detection rate of diseased targets, thereby improving detection completeness and exhibiting superior target recognition capabilities. To assess whether the proposed model suffers from overfitting, a comparative analysis was conducted between its performance on the training set and the test set. The experimental results revealed minimal discrepancies in both precision and recall, and the loss function exhibited smooth convergence throughout training. Notably, there was no observed trend of decreasing training error accompanied by increasing test error, which is a typical indicator of overfitting. These findings confirm that the performance improvements of the proposed model stem from its robust generalization capabilities rather than the memorization of the training data.

Figure 15 presents the comparative results between the original YOLOv10n model and the improved BED-YOLO model in detecting four types of tomato leaf diseases: early blight, late blight, septoria leaf spot, and leaf mold. The improved model demonstrated enhanced recognition performance across all four diseases, with improvements of 3.6% for early blight, 2.9% for late blight, 3.7% for septoria leaf spot, and 5.4% for leaf mold. The overall detection accuracy of the improved model significantly surpassed the original model, particularly in the detection of leaf mold. Leaf mold lesions are typically small, pale yellow or light green, with blurred edges and complex morphology, making them easily overlooked in natural environments. Moreover, the low contrast of the lesions often leads to weakened feature information due to background noise interference during feature extraction, increasing the risk of missed detections. The original YOLOv10 model struggled to accurately identify such diseases due to insufficient feature extraction capability and the loss of information on small targets in deep networks. In contrast, the improved BED-YOLO model excelled in detecting leaf mold by enhancing adaptability to morphological changes in the lesions. Even when the diseased areas were small or partially occluded, the model could accurately extract key features. Additionally, the optimized multi-scale feature fusion mechanism effectively retained information on small targets, thereby significantly improving the detection accuracy for leaf mold.

Figure 16 shows the confusion matrix, which further analyzes the performance of the improved model in classifying the four diseases. The results indicate that the improved model achieved higher recognition accuracy across all disease categories, demonstrating superior classification performance in various tasks.

### 4.4. Ablation Study

To comprehensively evaluate the contribution of each proposed module to overall model performance, a series of ablation experiments were conducted to assess the impact of different strategies on tomato leaf disease detection. All experiments were trained and tested on the same dataset, with improvement modules incrementally added to systematically examine their effects on detection accuracy. The experimental results are summarized in Table 2.

Group A served as the baseline experiment based on the original model. Group B introduced Deformable Convolutional Networks (DCNs) by replacing traditional convolutions in the YOLOv10 backbone with deformable convolutions to enhance the model’s sensitivity to diseased regions. The results showed that precision, recall, and mAP@50 increased by 1.1%, 1.2%, and 1.7%, respectively, indicating that DCN effectively improves feature extraction under complex conditions such as leaf occlusion and overlap. To investigate the independent contributions of BiFPN and EMA modules, Groups C and D evaluated these modules separately. Group C incorporated only the BiFPN structure into YOLOv10n to optimize multi-scale feature fusion. The experimental results demonstrated improvements of 0.5%, 0.6%, and 0.9% in precision, recall, and mAP, respectively, highlighting BiFPN’s capacity to enhance representation across scales, particularly for detecting small lesions. In Group D, only the Exponential Moving Average (EMA) attention mechanism was integrated into the neck structure to strengthen the model’s focus on critical regions. The results revealed increases of 0.6%, 0.7%, and 1.2% over the baseline, demonstrating that EMA effectively guides the model’s attention toward diseased areas while reducing background noise. Group E introduced BiFPN into the original FPN + PAN structure of YOLOv10n. This adjustment led to precision, recall, and mAP@50 improvements of 0.4%, 0.6%, and 1.3%, respectively, further validating BiFPN’s advantage in integrating multi-scale features and enhancing the recognition of fine-grained lesions. To assess the combined effects of different modules, Group F added the EMA attention mechanism to the DCN-enhanced model, forming the DCN + EMA configuration. Without BiFPN, this combination still achieved notable gains over Group B, with precision, recall, and mAP@50 increasing by 0.6%, 0.9%, and 1.6%, respectively. These results suggest that even without complex feature fusion structures, the EMA module can significantly suppress background interference and further boost detection accuracy by reinforcing attention to diseased regions. Finally, Group G represented the complete integration of all proposed modules, culminating in the final BED-YOLO model. This model achieved the best performance across all experimental groups, reaching a precision of 87.2%, a recall of 89.1%, and an mAP of 91.3%. The visual comparison of model performance across different ablation groups is illustrated in Figure 17. These results confirm that the synergistic combination of DCN, BiFPN, and EMA maximizes the model’s capacity for feature extraction, fusion, and attention focusing, making it particularly effective for small lesion detection in complex natural environments.

By progressively integrating DCN, BiFPN, and the EMA attention mechanism, the performance of the enhanced BED-YOLO model in tomato leaf disease detection was significantly improved. Each improvement module enhanced the model’s feature extraction and fusion capabilities from different aspects, allowing the model to achieve superior detection performance in complex natural environments.

### 4.5. Performance Comparison with Mainstream Models

To comprehensively evaluate the detection performance of the improved model, this experiment selected multiple state-of-the-art object detection models and conducted a comparative analysis with the proposed BED-YOLO model. The benchmark models included Faster R-CNN, YOLOv5n, YOLOv7-tiny, YOLOv8n, YOLOv9t, YOLOv11n, and the original YOLOv10n. To ensure a fair comparison across all models, training and evaluation were conducted on the same hardware platform using identical dataset partitions and training strategies. All YOLO-series models adopted their official baseline configuration files, with training parameters set to a batch size of 16, an initial learning rate of 0.01, and a total of 300 epochs using the SGD optimizer. Faster R-CNN was implemented using the MMDetection framework with ResNet-50 as the backbone. The input image size was uniformly resized to 640 × 640 to maintain consistency with the YOLO-series models. Key evaluation metrics included precision, recall, and mean average precision (mAP50), ensuring a holistic assessment of each model’s capability in tomato leaf disease detection.

Table 3 presents the detection results of different models. The data clearly indicate that the improved BED-YOLO achieved superior performance across all metrics, particularly excelling in mAP, precision, and recall. Firstly, regarding Faster R-CNN, as a two-stage object detection model, it demonstrated an advantage in object classification. However, due to its lower sensitivity to small objects when generating candidate regions, it struggled with the accurate detection of disease spots. Additionally, the high computational complexity and slow inference speed of Faster R-CNN made it unsuitable for the real-time detection of tomato diseases. Among the YOLO series models, YOLOv10n exhibited strong detection capabilities, achieving a precision of 85.1%, a recall of 86.2%, and an mAP50 of 87.4%. Moreover, YOLOv10n maintained a model size of 2.7 M parameters with floating-point operations (FLOPs) around 8.2 G, ensuring high precision while maintaining computational efficiency. Consequently, selecting YOLOv10n as the baseline model for further improvement was highly justified.

Both YOLOv8n and YOLOv11n also demonstrated competitive detection capabilities. Specifically, YOLOv8n attained a precision, recall, and mAP50 of 84.1%, 82.9%, and 86.9%, respectively. Its model size was approximately 3 M parameters, with FLOPs around 8.1G, which is comparable to YOLOv10n. However, both models slightly underperformed YOLOv10n, especially in complex background scenarios where detection stability was compromised. In contrast, YOLOv5n was a more lightweight model with only 2.5 M parameters and FLOPs of approximately 7.1 G. While computationally efficient, its detection performance was relatively lower, with a precision of 81.4%, a recall of 78.3%, and an mAP50 of 83.7%, making it the weakest among the YOLO series models evaluated. YOLOv7-tiny and YOLOv9t exhibited moderate detection performance. YOLOv7-tiny achieved a precision of 82.7%, a recall of 83.1%, and an mAP50 of 84.4%, slightly outperforming YOLOv5n but still trailing behind YOLOv8n. YOLOv9t recorded a precision of 83.8%, a recall of 80.5%, and an mAP50 of 86.1%, showing a marginal improvement over YOLOv7-tiny but delivering only moderate detection effectiveness.

The comparative analysis above demonstrates that the proposed enhanced model outperformed existing mainstream object detection frameworks in terms of key performance metrics such as precision, recall, and mAP, thereby confirming its effectiveness in the task of tomato leaf disease detection. Notably, under challenging conditions—such as complex natural backgrounds and small-object recognition—BED-YOLO exhibited superior robustness and discriminative capability, indicating substantial practical application potential. To further assess the model’s deployment feasibility, a comprehensive evaluation was conducted from the perspectives of architectural lightweightness and computational resource consumption. Compared to traditional two-stage detectors like Faster R-CNN, BED-YOLO offered a significant advantage in both parameter count and computational overhead, rendering it more suitable for deployment on resource-constrained embedded systems or edge computing devices. When benchmarked against lightweight single-stage models such as YOLOv5n and YOLOv8n, BED-YOLO maintained comparable inference speed while achieving higher detection accuracy and enhanced adaptability to variable environments. It is worth noting that while most existing studies emphasize improving detection accuracy, relatively few address the deployment challenges and practical usability of models in real-world agricultural scenarios. In contrast, this study prioritizes deployability as a core design principle from the outset, ensuring that the proposed model not only delivers strong detection performance but also meets the operational requirements of field-level implementation. In conclusion, BED-YOLO not only excels in experimental evaluation but also provides a robust technological foundation and practical roadmap for the real-world deployment of intelligent agricultural disease monitoring systems.

### 4.6. Analysis of Tomato Leaf Disease Detection Performance

To more intuitively assess the detection performance of the BED-YOLO model, this study compared the prediction results on the test set between the original YOLOv10n model and the improved BED-YOLO model. As shown in Figure 18, the left image displays the detection results of the original YOLOv10n model, while the right image shows the results of the improved BED-YOLO model. When using the YOLOv10n model, although the diseased leaves were detected, the confidence level was relatively low. In contrast, the improved BED-YOLO model not only accurately identified the diseased leaves but also exhibited a significant increase in confidence. This improvement can be attributed to the fact that, when detecting diseases in complex natural environments, factors such as lighting variations and natural background interference reduce the color and texture differences between the diseased regions and the surrounding environment, thereby increasing the detection difficulty. The original model is more susceptible to background noise, leading to low confidence levels and even false positives and missed detections. However, the BED-YOLO model can accurately locate the diseased regions even in complex backgrounds, thereby improving detection accuracy and significantly enhancing the stability and confidence of the predictions.

The results demonstrate that the improved BED-YOLO model outperforms the original model in natural environments, exhibiting superior performance in precise disease region recognition, target localization, and confidence enhancement.

## 5. Conclusions

In this study, we proposed an enhanced object detection model, BED-YOLO, based on YOLOv10n, to address the challenges of tomato leaf disease detection in complex natural environments. The intricate background, significant lighting variations, leaf occlusion, and small diseased regions pose considerable difficulties for traditional models. To overcome these challenges, we integrated Deformable Convolutional Networks (DCNs) into the backbone of YOLOv10n, replacing traditional convolutional layers to improve adaptability to occluded, overlapping, and blurred leaf lesions. Additionally, we incorporated a Bidirectional Feature Pyramid Network (BiFPN) into the FPN + PAN structure to enhance multi-scale feature fusion, thereby significantly improving the detection accuracy of small target lesions. Furthermore, the integration of the EMA mechanism within the C2f module effectively suppressed background noise and ensured the integrity and stability of lesion features, leading to improved overall detection performance. To validate the effectiveness of the proposed model, we compared BED-YOLO with mainstream object detection models, including YOLOv8n, YOLOv5n, and Faster R-CNN. The experimental results demonstrated that the improved model outperformed others in terms of precision, recall, and mean average precision (mAP), proving its superiority in tomato leaf disease detection tasks. Moreover, comparative analysis between the original YOLOv10n model and the improved BED-YOLO model on four types of tomato leaf diseases—early blight, late blight, septoria leaf spot, and leaf mold—revealed significant improvements in detection accuracy. When detecting lesions on the same tomato leaf image, the original model exhibited lower confidence, while the improved BED-YOLO model accurately localized the diseased areas with higher confidence and greater stability.

## 6. Future Prospects

Despite the promising performance of the enhanced BED-YOLO model in tomato leaf disease detection, several challenges remain. First, the evolving agricultural production environment introduces more complex and diverse disease types and lesion patterns, requiring further improvements in the model’s generalization capability. In particular, enhancing adaptability across different regions and tomato varieties is essential. Additionally, the scale and quality of datasets are critical for the performance of object detection models. However, agricultural disease datasets often suffer from limited sample sizes and high annotation costs. Efficient training on limited datasets while avoiding overfitting remains a challenge for future research. Furthermore, the lightweight nature and real-time performance of the model are key bottlenecks for the widespread application of intelligent agricultural detection systems. Balancing high detection accuracy with reduced computational complexity and achieving seamless deployment on edge devices will be the focus of future research. We aim to explore multi-source data fusion, transfer learning, and weakly supervised learning methods to enhance model robustness and applicability, thereby advancing intelligent agricultural disease detection technology.

## Figures and Tables

**Figure 1 sensors-25-02882-f001:**
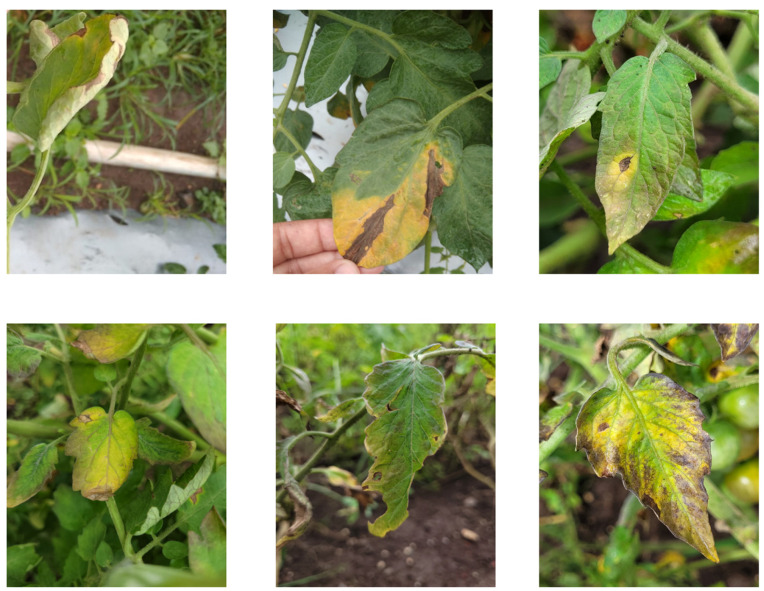
Sample images of various diseases.

**Figure 2 sensors-25-02882-f002:**
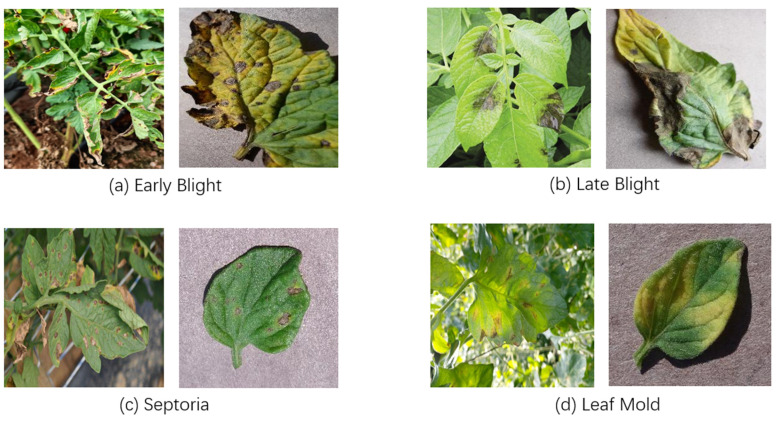
Disease image data samples.

**Figure 3 sensors-25-02882-f003:**
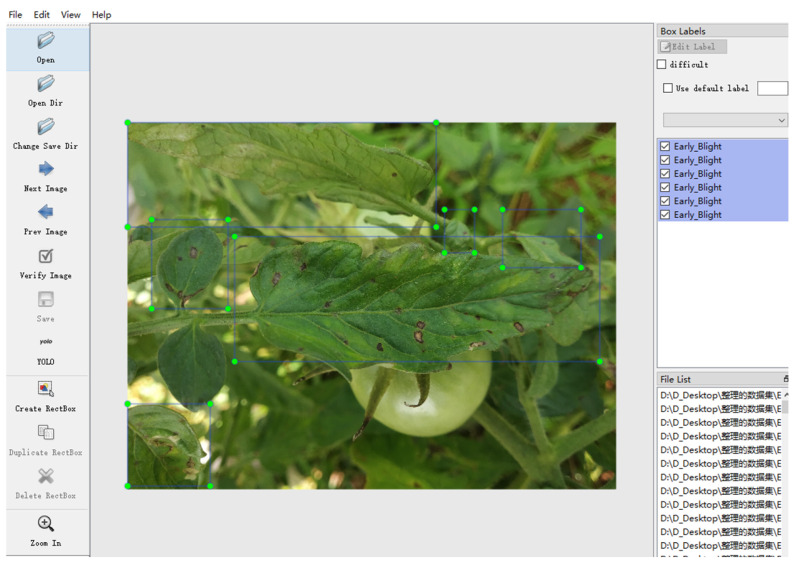
Example of disease labeling data.

**Figure 4 sensors-25-02882-f004:**
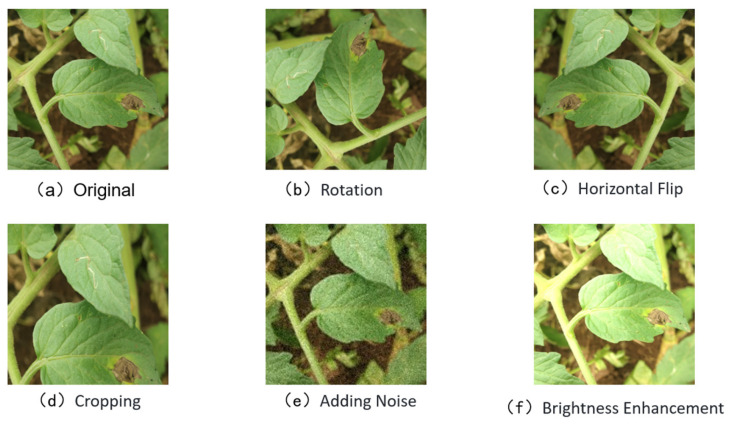
Examples of data enhancement.

**Figure 5 sensors-25-02882-f005:**
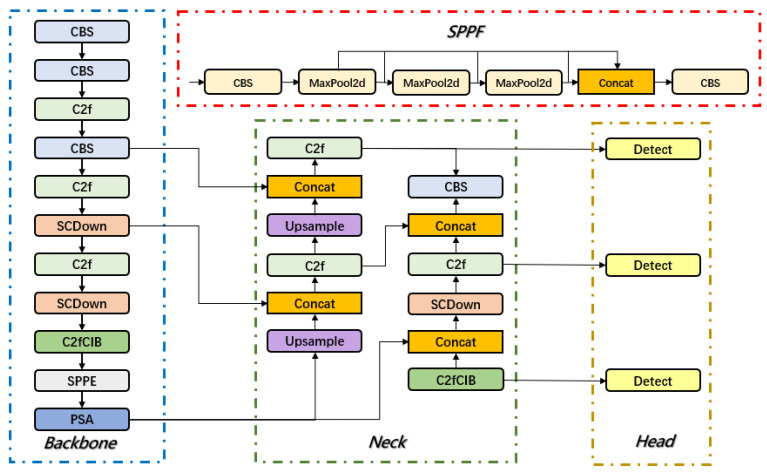
YOLOv10 network architecture.

**Figure 6 sensors-25-02882-f006:**
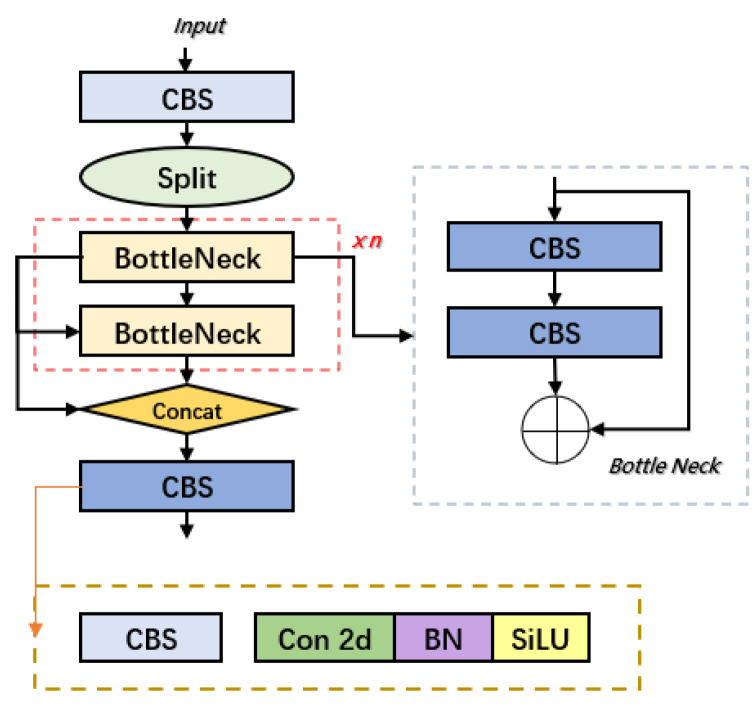
C2f network architecture.

**Figure 7 sensors-25-02882-f007:**
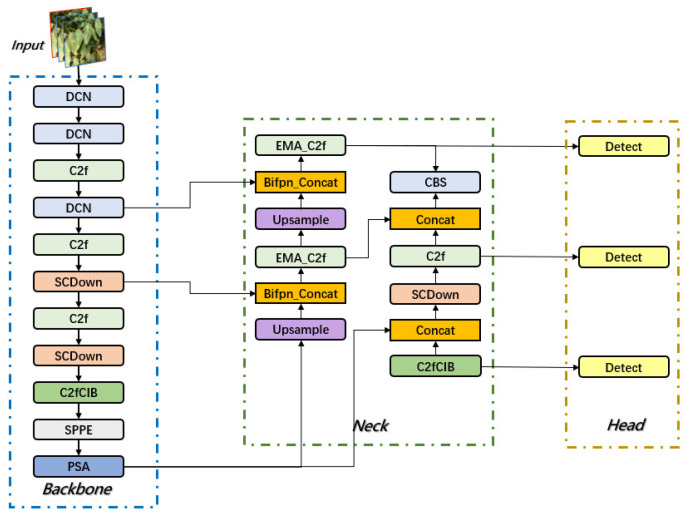
BED-YOLO network architecture.

**Figure 8 sensors-25-02882-f008:**
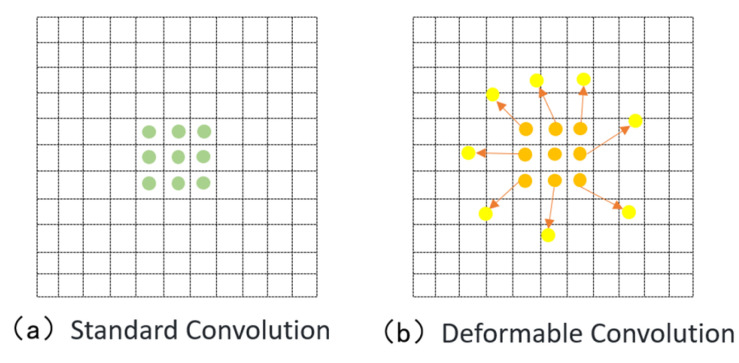
The difference between standard convolution and deformable convolution.

**Figure 9 sensors-25-02882-f009:**
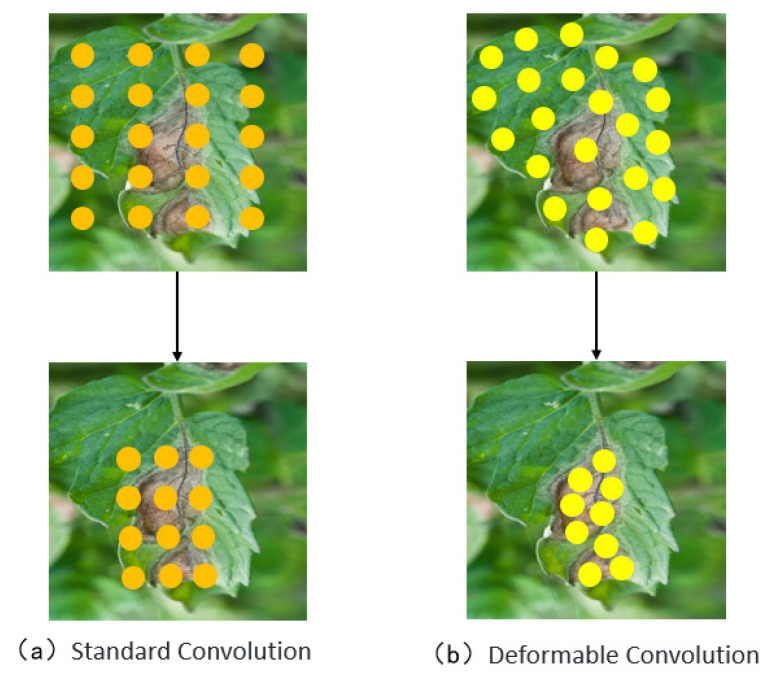
Flexibility of deformable convolutions.

**Figure 10 sensors-25-02882-f010:**
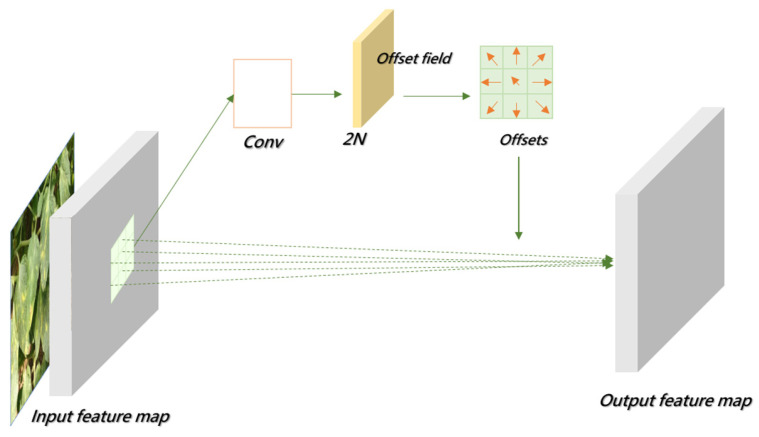
DCN architecture.

**Figure 11 sensors-25-02882-f011:**
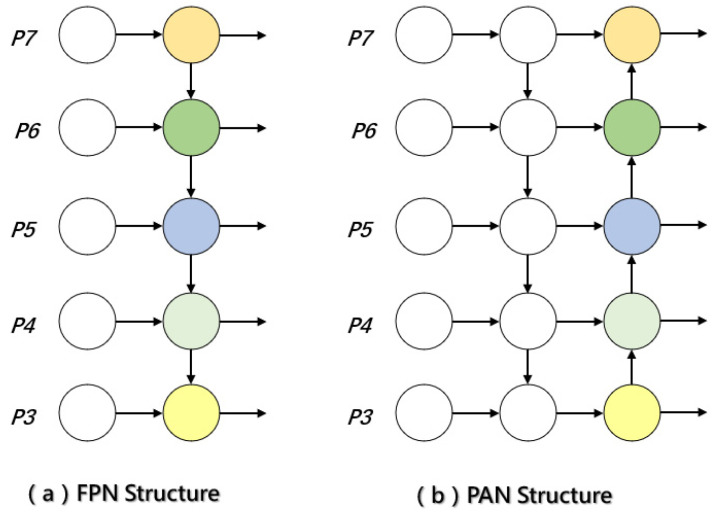
FPN and PAN architecture.

**Figure 12 sensors-25-02882-f012:**
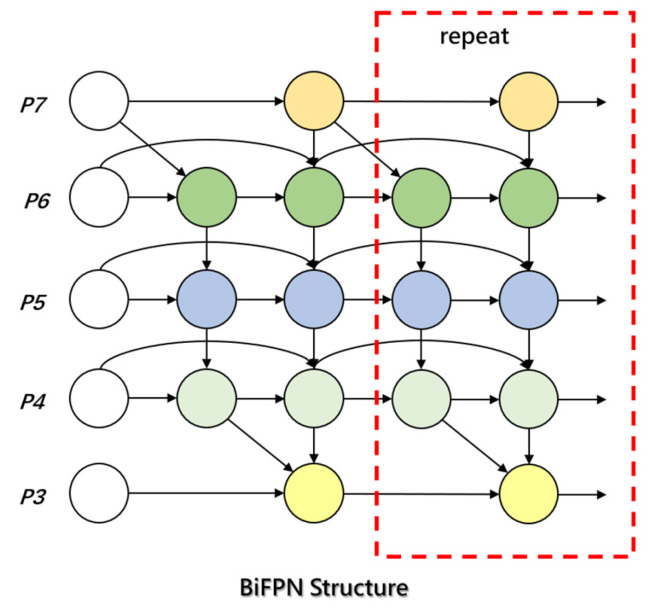
BiFPN architecture.

**Figure 13 sensors-25-02882-f013:**
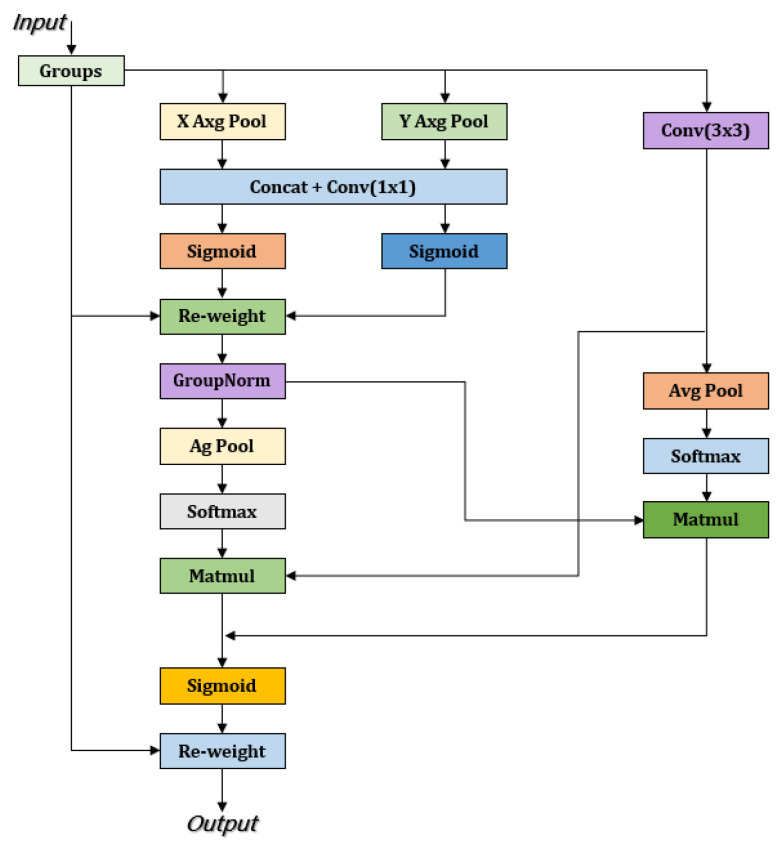
EMA network architecture.

**Figure 14 sensors-25-02882-f014:**
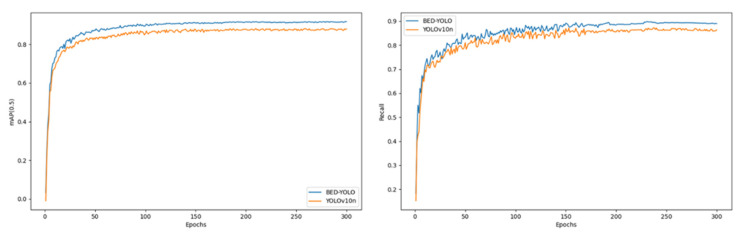
Comparison of BED-YOLO visualization results.

**Figure 15 sensors-25-02882-f015:**
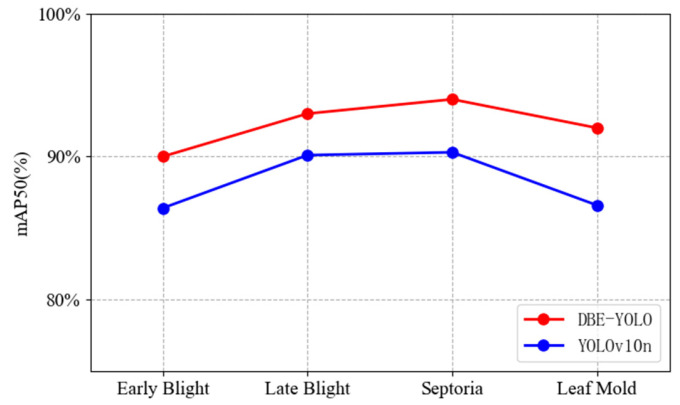
Comparison of disease detection effect.

**Figure 16 sensors-25-02882-f016:**
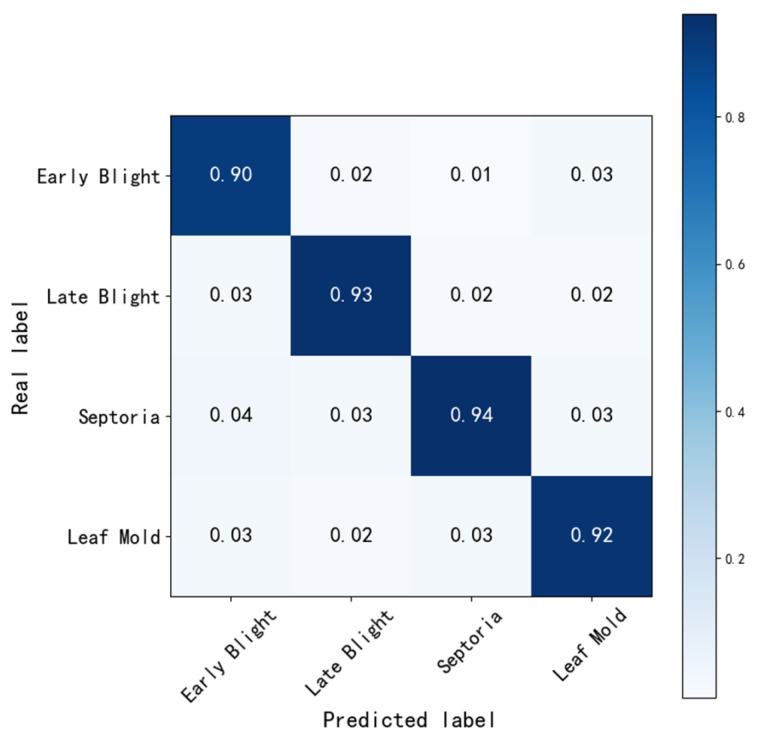
Detection performance of different diseases.

**Figure 17 sensors-25-02882-f017:**
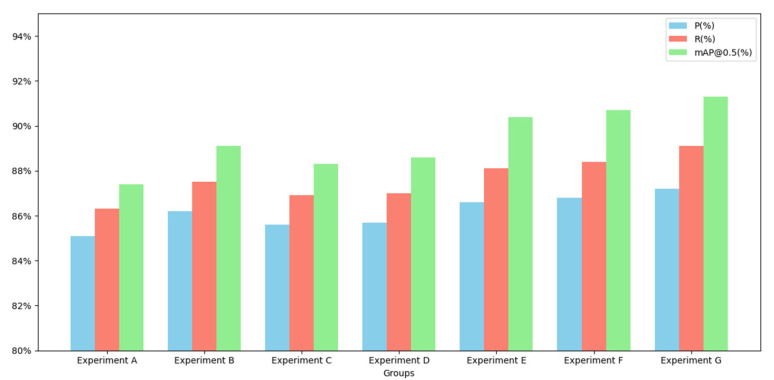
Comparison of different experimental groups.

**Figure 18 sensors-25-02882-f018:**
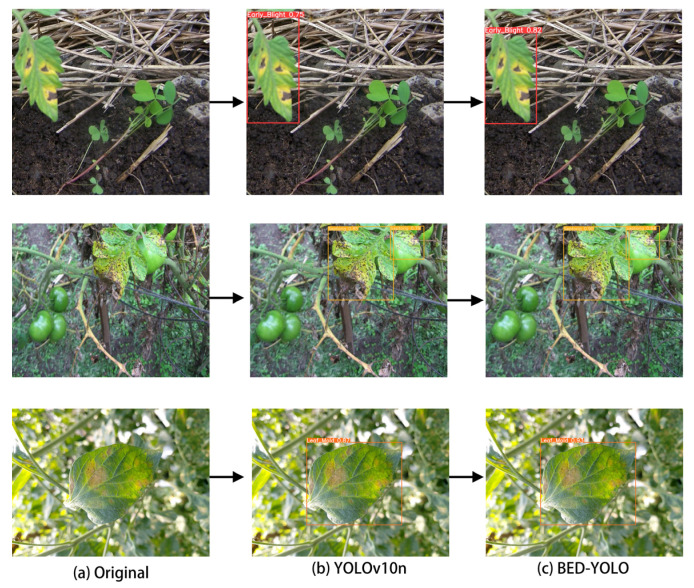
Comparison of BED-YOLO prediction results.

**Table 1 sensors-25-02882-t001:** Environment configuration.

Name	Environmental Parameters
Operating System	Windows10
GPU	NVIDIA GeForce RTX 4060
CPU	IntelCorei5-12400F
Python	3.11
Pytorch	2.0.0
CUDA	11.8

**Table 2 sensors-25-02882-t002:** Ablation experiment results.

	DCN	BiFPN	EMA	P (%)	R (%)	mAP	Params/M	Flops/G
A				85.1	86.3	87.4	2.3	6.7
B	√			86.2	87.5	89.1	2.8	7.9
C		√		85.6	86.9	88.3	2.6	7.4
D			√	85.7	87.0	88.6	2.5	7.2
E	√	√		86.6	88.1	90.4	3.4	9.2
F	√		√	86.8	88.4	90.7	3.1	8.7
G	√	√	√	87.2	89.1	91.3	3.7	9.8

**Table 3 sensors-25-02882-t003:** Test results of different models.

	P (%)	R (%)	mAP	Params/M	Flops/G
Faster R-CNN	60.2	62.3	65.4	137.0	370
YOLOv5n	81.4	78.3	83.7	2.5	7.1
YOLOv7-tiny	82.7	83.1	84.4	6.3	13.2
YOLOv8n	84.1	82.9	86.9	3.0	8.1
YOLOv9t	83.8	80.5	86.1	1.9	7.6
YOLOv10n	85.1	86.2	87.4	2.7	8.2
YOLOv11n	83.5	82.3	86.6	2.6	6.3
BED-YOLO	87.2	89.1	91.3	3.7	9.8

## Data Availability

The data involved in the study can be obtained by contacting the authors.

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
