# Peer review of "BED-YOLO: An Enhanced YOLOv10n-Based Tomato Leaf Disease Detection Algorithm"

_sensors, 2025, doi:10.3390/s25092882_

Round 1
Reviewer 1 Report
Comments and Suggestions for Authors
1. In the introduction, it would be beneficial to add some transition sentences that explain the advantages or motivations for applying the proposed method, supported by relevant citations.
2. Please clarify the ratio between the two datasets. Are the different data types distributed evenly?
3. Provide a brief description of the annotation process, rather than relying solely on an image.
4. It would be helpful to include more details about YOLO v10 and NMS, explaining how they function. Adding more citations would also strengthen the section.
5. Please expand the explanation for Figure 7 to highlight the differences in principle between standard convolution and variable convolution.
6. Clarify what is meant by “offset value” in the statement: “The principle of deformable convolution relies on a network that learns offset values.”
7. Expand the description of deformable convolution: “Based on the original formula, deformable convolution introduces an offset for each sampling point, generated by an additional convolution layer from the input feature map, typically represented as a fractional value.” — More detail would make this clearer.
8. Please elaborate on the principles behind FPN and PAN, explaining how they work and why they enhance performance: for example, how FPN aids in extracting multi-scale feature information and how PAN improves information flow across layers, particularly for small object detection.
9. There is a typo in the legend of Figure 13 that needs correction.
Reviewer 2 Report
Comments and Suggestions for Authors
1.It is recommended that the authors clarify whether the model has been evaluated for overfitting. Specifically, it should be examined whether the reported performance improvements might be partially or entirely attributed to overfitting.
2.It would strengthen the study if the authors could compare the model’s performance on datasets with simple versus complex backgrounds. Analyzing the performance gap between these scenarios could provide deeper insights into the model’s effectiveness under real-world conditions.
3.The reproducibility of the proposed approach is a critical concern. The authors are advised to state whether the dataset used in the experiments is publicly available, thereby enabling independent verification of the results.
4.While the paper investigates the combined use of DCN, BiFPN, and EMA, it does not explore the effect of using DCN and EMA in isolation from BiFPN. An ablation study that includes this combination would help determine whether this subset configuration could further improve performance.
5.The manuscript presents a comparative analysis of several models (Faster R-CNN, YOLOv5n, YOLOv7-tiny, YOLOv8n, YOLOv9t, and YOLOv10n) on the same dataset. However, the specific experimental settings and parameter configurations are not sufficiently detailed. Providing this information is essential to ensure the transparency and reproducibility of the experiments.
Reviewer 3 Report
Comments and Suggestions for Authors
The presented paper proposes an improved YOLOv10n architecture for detecting diseases on tomato leaves. While the structure of the research is clear, there are several significant weaknesses that should be addressed:
The term DBE-YOLO, as well as several of the proposed improvements to YOLO, have already been introduced in Zhou et al. https://www.mdpi.com/2075-5309/15/7/1117. The authors should consider using a different term to avoid confusion. Additionally, they should reference Zhou et al.'s paper, explain its relevance, and clearly highlight the conceptual differences between the proposed method and existing approaches.
The research topic “plant disease detection” is widely studied, and numerous review papers are published annually. While the authors mention some related studies, they should include a brief overview of more recent works, especially those specifically focused on tomato disease detection from 2022 to 2025. This would help situate the current work within the context of existing literature and emphasize its novelty.
The dataset used in this study consists of two parts: a self-collected dataset and a subset of the PlantVillage dataset. Although PlantVillage is commonly used for classification tasks, it was collected under controlled conditions and its applicability to real-world scenarios is limited. Moreover, its use for object detection—particularly for detecting diseased leaves rather than specific lesions—is questionable due to its low resolution (256×256) and the fact that it contains only isolated tomato leaves. The authors should provide detailed information about their self-collected dataset, including the number of images, and justify the inclusion of PlantVillage data in the context of object detection.
Section 2.2 (Data Annotation and Augmentation) contains basic information that may be considered self-evident. This section could be merged into Section 2.1, as the content does not warrant a separate section.
While the experimental results show improvements in YOLO performance metrics, the lack of comparison with other state-of-the-art methods makes it difficult to assess the significance of the proposed modifications. It is recommended that the authors evaluate their approach on a recognized benchmark dataset to facilitate fair comparisons with recent methods. Furthermore, including more visual examples to illustrate the effectiveness of the proposed model would enhance the credibility and clarity of the results.
Reviewer 4 Report
Comments and Suggestions for Authors
Dear Authors,
The manuscript titled “DBE-YOLO: An Enhanced YOLOv10n-Based Tomato Leaf Disease Detection Algorithm” addresses an important problem in the domain of tomato disease identification using convolutional neural networks. The paper is generally well-written and easy to follow. However, it currently lacks critical details necessary for publication.
While the proposed modifications may contribute to improving detection accuracy, the current structure and content of the manuscript do not offer sufficient novelty or insights that would be of significant interest to the research community. As many existing studies already focus on modifying YOLO-based architectures for accuracy improvements, it is essential to go beyond architectural changes. Specifically, only a limited number of works have taken the crucial step of deploying their models on mobile or edge devices to evaluate performance and limitations in real-world scenarios.
To enhance the impact and relevance of the work, I strongly recommend a major revision. This should include a more comprehensive comparative analysis with existing literature, practical deployment results, and a clearer demonstration of how the proposed method advances the state of the art.
Abstract:
More description regarding the dataset is needed. There is no information about what diseases were evaluated, the class proportions, how many instances were annotated, bounding boxes, or segmentation masks, etc.
This statement, “ making it more suitable for large-scale, real-world tomato leaf disease detection tasks,” needs clarification since there is no information about whether the evaluation was conducted in field or laboratory conditions.
Introduction:
More explanation is needed to justify the reason for choosing the YOLOv10 model. Yolov10 does not have NMS, which makes the identification of small objects more challenging. Why didn't you use Yolov11? Yolov11 has significant improvements compared to the previous models.
Material and methods:
Please describe the image resolution and what kind of camera was used for data collection.
Why did the authors include the PlantVillage dataset? Introducing various diseases from different plants can challenge the model's ability to identify similar objects. Moreover, if the public dataset is not validated by an expert, the disease identification may be compromised.
Please check for spelling words. Line 344 should be Precision
Why was the comparison of the proposed model performed only with the YOLOv10n model? There are many manuscripts that modify YOLO-based models without conducting an in-depth comparative analysis across different model sizes and versions.
Please provide a more in-depth discussion of the results in comparison with relevant literature. As previously mentioned, while numerous studies have proposed modifications to the YOLO architecture to enhance detection accuracy, only a limited number have gone further by deploying their models in real-world scenarios, such as on mobile or edge devices, to evaluate practical performance and limitations. In its current form, the manuscript lacks novel insights and does not sufficiently engage readers or contribute meaningfully to the existing body of knowledge. A more thorough analysis and discussion would significantly strengthen the manuscript’s impact and relevance.
Round 2
Reviewer 2 Report
Comments and Suggestions for Authors
-
The current module combination strategy appears to overlook the individual contributions of BiFPN and EMA, focusing solely on the impact of the DCN module. Would the inclusion of experiments evaluating BiFPN and EMA independently—alongside all possible combinations—strengthen the persuasiveness and completeness of the study?
-
Given that the original model is more susceptible to background noise, while BED-YOLO demonstrates improved robustness, would it be more convincing and scientifically rigorous to conduct experiments on a subset of data with high background noise to substantiate this claim?
Reviewer 3 Report
Comments and Suggestions for Authors
The authors have done commendable work; however, there are several aspects that require further clarification and improvement:
- Lines 117–127: Citations supporting the claims made in this section are missing. Relevant references should be provided to validate the presented statements.
- Dataset Description: The information provided about the self-collected dataset is insufficient. The authors should include the class distribution of the samples and provide a variety of visual examples illustrating the following:
“Additionally, the dataset includes leaf images from different stages of disease progression, such as early-stage lesions, mid-stage infections, and severely infected leaves. Furthermore, images containing backgrounds like soil, weeds, and other vegetation were incorporated to enhance the model's adaptability to complex environments.”
Readers need to clearly understand the characteristics of the dataset being used. The phrase “approximately 65%” is vague; a complete breakdown of the dataset is necessary. Details should be provided for both the self-collected dataset and the additional data obtained from PlantVillage. The data augmentation process also requires clarification. It appears that the authors generated augmented samples, rather than using online augmentation techniques such as torchvision.transforms in PyTorch. Therefore, it is important to specify the original data distribution, which classes were augmented and resized, and what exact augmentation techniques were applied.
- Lines 274–276:
“These fundamental differences in application scenarios and design objectives underscore the novelty and distinctiveness of our proposed approach.”
While differences between crack detection and plant disease lesion detection do exist, referring to them as "fundamental" may be overstated. It would be advisable to use more moderate language here to avoid sounding overly assertive.
- Use of the PlantVillage Dataset:
The rationale behind including the PlantVillage dataset remains unclear. The objective appears to be detecting disease lesions on leaves, whereas PlantVillage images consist of single, isolated leaves. The authors should justify the inclusion of this dataset more clearly.
The model should be trained on the self-collected dataset alone, as well as on a combined dataset including PlantVillage data.
The evaluation should then be performed separately on:
The self-collected dataset
The PlantVillage dataset
A mixed evaluation set (self-collected + PlantVillage)
This would demonstrate whether including PlantVillage data offers any tangible benefit to the model's performance in real-world scenarios as well as effectiveness of the authors YOLO modification method.
Reviewer 4 Report
Comments and Suggestions for Authors
The revised manuscript does not present substantial improvements that would justify its publication. As pointed out during the initial review, numerous existing studies already propose modifications to YOLO and other CNN-based models for specific tasks, often claiming superiority over standard models. However, these claims frequently include statements about potential deployment on edge or mobile devices, yet, in this case, such deployment has not been tested or demonstrated.
The authors have revised the introduction to suggest that YOLOv10 is superior due to the absence of NMS; however, this claim is inaccurate. Furthermore, the paragraph stating that “YOLOv11 has demonstrated strong performance in general object detection benchmarks, [but] it lacks extensive validation in plant disease scenarios...” is not only speculative but also unsupported by any citations. The assertion that its design may compress critical information when detecting small lesions is unsubstantiated and misleading.
While some corrections have improved the clarity of certain sections, the authors have not extended their validation to real-world field scenarios, nor have they tested the model’s applicability or limitations on actual edge devices.
Considering these issues and the limited interest this manuscript offers to the readership, I recommend that it be rejected.
Round 3
Reviewer 3 Report
Comments and Suggestions for Authors
The authors have taken into account several of my suggestions and comments and have provided a justification for not implementing the proposal to test their methods on various data configurations. I have no further suggestions for the authors.
Reviewer 4 Report
Comments and Suggestions for Authors
Dear authors,
Thank you for sharing the corrected manuscript version and explaining my questions. I think the manuscript looks better and can be published.